# Ecotype-Specific and Correlated Seasonal Responses of Biomass Production, Non-Structural Carbohydrates, and Fatty Acids in *Zostera marina*

**DOI:** 10.3390/plants13030396

**Published:** 2024-01-29

**Authors:** Pedro Beca-Carretero, Clara Marín, Tomás Azcárate-García, Claudia L. Cara, Fernando Brun, Dagmar B. Stengel

**Affiliations:** 1Botany and Plant Science, School of Natural Sciences, University of Galway, H91 TK33 Galway, Irelanddagmar.stengel@universityofgalway.ie (D.B.S.); 2Department of Theoretical Ecology and Modelling, Leibniz Centre for Tropical Marine Research, 28359 Bremen, Germany; 3Centro de Investigación Marina, Facultad de Ciencias del Mar, Universidad de Vigo, 36310 Vigo, Spain; claramarinperez@gmail.com; 4Department of Evolutionary Biology, Ecology and Environmental Sciences & Biodiversity Research Institute (IRBio), University of Barcelona, 08028 Barcelona, Spain; azcarate.garcia.t@gmail.com; 5Department of Marine Biology and Oceanography, Institute of Marine Sciences (ICM-CSIC), 08003 Barcelona, Spain; 6Department of Biology, Division of Ecology, Faculty of Marine and Environmental Sciences, University of Cadiz, 11510 Puerto Real, Spain; fernando.brun@uca.es

**Keywords:** seagrasses, Ireland, morphometric descriptors, fatty acids, non-structural carbohydrates, productivity, ecological indicator, eco-physiological tool, seasonality

## Abstract

Seagrasses, which are marine flowering plants, provide numerous ecological services and goods. *Zostera marina* is the most widely distributed seagrass in temperate regions of the northern hemisphere, tolerant of a wide range of environmental conditions. This study aimed to (i) examine seasonal trends and correlations between key seagrass traits such as biomass production and biochemical composition, and (ii) compare seasonal adaptation of two ecotypes of *Z. marina* exposed to similar environmental conditions on the west coast of Ireland. During summer, plants accumulated higher levels of energetic compounds and levels of unsaturated fatty acids (FAs) decreased. Conversely, the opposite trend was observed during colder months. These findings indicate a positive seasonal correlation between the production of non-structural carbohydrates and saturated fatty acids (SFAs), suggesting that seagrasses accumulate and utilize both energetic compounds simultaneously during favorable and unfavorable environmental conditions. The two ecotypes displayed differential seasonal responses by adjusting plant morphology and production, the utilization of energetic reserves, and modulating unsaturation levels of fatty acids in seagrass leaves. These results underscore the correlated seasonal responses of key compounds, capturing ecotype-specific environmental adaptations and ecological strategies, emphasizing the robust utility of these traits as a valuable eco-physiological tool.

## 1. Introduction

Global change stressors, such as climate change, pollution, and habitat destruction, have been identified as major drivers of ecosystem alterations on Earth [1,2]. These stressors are causing significant impacts on biodiversity, including species loss and decline, and disruptions in ecosystem functioning [3,4]. To assess their detrimental effects on key ecosystems, the development of ecological and physiological tools has become invaluable [5]. For example, the analysis of selected biochemical compounds can provide insights into plant adaptation and acclimation mechanisms in response to current environmental change [6,7].

Seagrasses are flowering plants in shallow and sheltered marine areas of intertidal and subtidal zones [8,9]. Their meadows represent important ecosystems with significant ecological value, serving as wave energy dissipaters, water filters, sediment anchors, and nurseries for fish and shellfish [10,11,12]. Seagrasses also play a crucial role as a carbon sink, thus helping to mitigate climate change impacts [13]. However, despite their importance, seagrasses are among the most endangered ecosystems worldwide, with global habitat extent losses of 7% per year [14]. Seagrasses face numerous global and local threats, including climate change, nutrient and sediment runoff, and coastal development that degrade water quality, leading to die-backs and habitat loss [15]. Despite these challenges, there have been encouraging signs of local recovery due to conservation and management efforts in recent decades [16,17].

*Zostera marina* is the seagrass species with the widest latitudinal range in the northern hemisphere [16]. Its high level of phenotypic, morphological, and physiological adaptability enables this species to thrive in diverse environmental conditions and habitats [18]. Growth, photosynthesis, and reproduction of seagrasses are primarily affected by irradiance, temperature, and nutrient availability [19]. *Zostera marina* typically exhibits distinct seasonal patterns, with maximal growth rates in spring and summer when temperatures and irradiances are high, and reduced biomass production and photosynthetic activity in autumn and winter under less favorable conditions [20,21]. Temperate seagrasses display various physiological and biochemical adjustments during the seasonal cycle [22], including modifications in plant morphology and production, utilization of energetic reserves, and the modulation of physicochemical properties of key functional structures such as the membrane of thylakoids in photosynthetic tissues [23,24,25]. Non-structural carbohydrates (NSC), such as sucrose and starch, are crucial energetic compounds for optimal plant growth and physiological functioning, with sucrose being the main energy sources in seagrasses [26,27]. Seagrasses usually accumulate the highest amounts of NSC during summer–autumn, when environmental conditions for photosynthesis are more favorable, while under suboptimal conditions, such as in winter, energetic reserves are utilized [28,29,30].

Fatty acids (FAs) are major components of plant membranes and also play a critical role in energy storage and signaling processes [31]. FAs can be categorized into specific groups based on their structural and biochemical properties [32,33]. Monounsaturated fatty acids (MUFAs) have a primarily structural function in the membranes of organelles (i.e., Golgi apparatus, endoplasmic reticules) [34]. Polyunsaturated fatty acids (PUFAs) are incorporated into structural compounds of chloroplasts, promoting membrane fluidity [35,36,37]. Saturated fatty acids (SFAs) are also structural compounds, providing membrane stability and functionality, and are key energetic compounds, usually accumulated as triacylglycerols (TAGs) [38,39]. FAs composition of seagrass photosynthetic tissues was identified as a sensitive environmental indicator [40,41,42], which was correlated with variation in temperature, irradiance, and nutrient concentration. In particular, *n*-3 PUFAs and unsaturation indicators decrease in response to increases in temperature and nutrient stress [32,35]. Moreover, seagrasses synthesize essential omega-3 and omega-6 PUFAs which are subsequently available for higher trophic levels. These compounds are critical in regulating physiological and biological functions of consumers, such as reproduction, vision, or neuronal transmission [43,44,45].

In Ireland, seagrass meadows cover an estimated 67 km^2^, with *Zostera marina* and *Zostera noltei* being the most abundant species [37,41,46,47]. The common ecotype of *Zostera marina* is typically distributed in the subtidal zone, exhibiting perennial behavior and developing large photosynthetic structures. By contrast, *Zostera marina* var. angustifolia is an ecotype of *Z. marina* that is typically found in intertidal areas along the east coast of Ireland [48,49,50], characterized by smaller-sized individuals. Ecotypes are populations or subspecies adapted to local environmental conditions defined by specific physiological, morphological, and ecological attributes [51,52,53]. In contrast to the perennial nature of common *Z. marina* plants, *Z. marina* var. angustifolia exhibits an apparent annual growth pattern, disappearing during unfavorable conditions in winter and re-establishing itself in warmer months [54]. Along the coast of Ireland, *Zostera marina* angustifolia is commonly found in association with *Z. noltei* but not in interaction with perennial populations of *Z. marina*. In this context, the cohabitation of perennial and annual ecotypes of *Z. marina* has not been previously documented in Ireland. The overall health of Irish seagrass meadows is considered good but some intertidal populations, mostly located on the east and south coast, may be threatened by anthropogenic pressures [37,41,47].

The relationship between biomass production, carbohydrates, and fatty acid accumulation and synthesis during an annual cycle provides crucial insights into seagrass capacity to adapt and acclimate to different environmental gradients. However, there is limited research into a potential correlation between these compounds in seagrasses. Therefore, this study aimed to address this gap by (i) investigating seasonal cycles of key seagrass traits, including production of photosynthetic structures, non-structural carbohydrate concentrations and fatty acid level and composition; (ii) examining potential seasonal correlations between seagrass descriptors; and finally, (iii) comparing seasonal acclimation strategies (plant development and biochemical compounds) of the two *Z. marina* ecotypes exposed to similar environmental conditions.

We hypothesize that the seasonal production and accumulation of target biochemical compounds, including non-structural carbohydrates and fatty acids, are likely influenced by changes in environmental variables such as temperature and irradiance. Furthermore, we expect a seasonally varying correlation between descriptors due to their functional and physiological interdependence and acclimation to different environmental gradients.

## 2. Results

### 2.1. Environmental Descriptors

Throughout the study period, both seawater surface temperature (SST) and irradiance displayed a consistent seasonal pattern, with minimum values in winter and maximum values in summer. SST reached its minimum in March (6.76 ± 0.31 °C), followed by a rapid increase to peak temperatures in July (16.67 ± 0.41 °C). Intermediate SST values were recorded in November–December and April–May (9–12 °C). On the other hand, minimum irradiance levels were observed in December (473.64 ± 173.11 Wh m^−2^), with the highest values in June (5823.92 ± 1920.06 Wh m^−2^). Intermediate values were recorded in March–April and August–September (2400–3800 Wh m^−2^) (Figure 1).

### 2.2. Seasonal and Ecotype-Specific Responses of Shoot Leaf Area, Biomass Production, and Leaf Area Index

In this study, we characterized two *Z. marina* ecotypes: one representing the perennial *Z. marina* ecotype, identified as ZM1, and the second ecotype, representing the annual and smaller *Z. marina* ecotype known as *Z. marina* angustifolia, designated as ZM2 (further details are outlined in Section 4). All seagrass descriptors displayed significant differences across the factors of ‘month’ and ‘ecotype’, and interactions between these two factors were also significant (Table 1). Overall, ZM1 showed significantly higher values in leaf area, biomass production, and leaf area index (LAI) than ZM2 (PERMANOVA, *p* < 0.05) (Table 1). The shoot leaf area, biomass production, and LAI reported their maximum values in warmer months (August–September) and the lowest in colder months (January–February), which coincided with the minimum annual temperature and irradiance levels at this time of the year (Figure 2). For instance, total leaf area of both ecotypes was highest in September (ZM1 = 15.46 ± 7.23 cm^2^ shoot^−1^; ZM2 = 5.42 ± 2.28 cm^2^ shoot^−1^) and minimal in February (ZM1 = 1.66 ± 0.97 cm^2^ shoot^−1^; ZM2 = 0.25 ± 0.10 cm^2^ shoot^−1^) (Figure 2A). Similarly, for both ecotypes, maximum leaf biomass production rates were observed in September (ZM1 = 0.0021 ± 0.00047 g DW d^−1^ shoot^−1^; ZM2 = 0.0011 ± 0.0003 g DW d d^−1^ shoot^−1^) and minimum rates in February (ZM1 = 0.00054 ± 0.00014 g DW d^−1^ shoot^−1^) and January (ZM1 = 0.00044 ± 0.00003 g DW d^−1^ shoot^−1^) (Appendix A).

Shoot leaf area, leaf biomass production, and LAI of both ecotypes were significantly affected by the interaction between ‘month’ and ‘ecotype’. For instance, leaf biomass production of ZM1 showed a progressive increase from its minimum in February until its maximum in summer. However, after starting from a minimum in January, shoot leaf area leaf and biomass production of ZM2 was delayed until May; at this time, the vegetative growth phase started and a maximum in leaf biomass was then observed in September (Figure 2B). Another significant difference between the seasonal pattern of the two ecotypes was the fact that ZM1 maintained maximum LAI from July to September, whereas LAI in ZM2 displayed very sharp rises from minimum values in April to a maximum in September, followed then by marked sharp declines until October (Figure 2C, Appendix A).

### 2.3. Seasonal and Ecotype-Specific Responses of Biochemical Compounds

#### 2.3.1. Sucrose and Starch Levels

Sucrose and starch levels in seagrass leaves varied significantly according to the factors of ‘month’ and ‘ecotype’, and the interaction between these two factors was also significant (PERMANOVA, *p* < 0.05) (Figure 3, Table 2). Both ecotypes accumulated significantly more sucrose by August (summer) (ZM1 = 135.78 ± 20.10 mg g^−1^ DW; ZM2 = 173.29 ± 4.12 mg g^−1^ DW) (Appendix A), and plants contained significantly less sucrose in winter (PERMANOVA, *p* < 0.05). Particularly, ZM2 (ZM2 = 6.94 ± 2.88 mg g^−1^ DW) had the lowest contents of sucrose in January, which were 61% lower than those of ZM1. Average monthly starch levels in ZM1 (55.87 ± 6.68 mg g^−1^ DW) were significantly higher than those in ZM2 (29.01 ± 6.47 mg g^−1^ DW). The highest concentration of starch in ZM1 was observed in September (72.10 ± 3.50 mg g^−1^ DW), while for ZM2, it was reached in April (43.43 ± 5.77 mg g^−1^ DW) (Appendix A).

#### 2.3.2. Fatty Acids Content and Composition

No significant differences in total fatty acids (TFAs) content were found between the two ecotypes (ZM1 = 2.2 ± 0.2% of DW; ZM2 = 2.1 ± 0.2% of DF) (Appendix A). For both, higher TFA contents were observed in spring (ZM1 = 3.03 ± 0.054% of DW; ZM2 = 2.83 ± 0.2% of DW), while lowest values (ZM1 = 1.68% ± 0.11 of DW; ZM2 = 1.71 ± 0.16% of DW) were reached in August–September (Figure 4). Overall, PUFA/SFA, *n*-3/*n*-6 PUFA and the unsaturation index 18:3 *n*-3/16:0 displayed significant differences between the factors of ‘month’ and ‘ecotype’, and the interaction between these two factors was also significant (PERMANOVA, *p* < 0.05) (Table 3). Specifically, PUFA/SFA showed a strong seasonal trend, with maximum ratios observed in spring (ZM1 = 3.83 ± 0.03; ZM2 = 3.01 ± 0.06) and the lowest ratios in summer (ZM1 = 2.65 ± 0.05; ZM2 = 2.40 ± 0.02), coinciding with the warmest months (Appendix A). Similarly, omega-3/6 displayed a significant seasonal trend, with maximum ratios in spring (ZM1 = 5.47 ± 0.11; ZM2 = 5.8 ± 0.13) and minimum ratios in summer (ZM1 = 3.01 ± 0.17; ZM2 = 3.4 ± 0.09) (Figure 5). However, regarding 18:3 *n*-3/16:0, both ecotypes showed different seasonal responses; ZM1 reached maximum values in the coldest months (January–February), whereas ZM2 contained maximum levels in April–May. Both ecotypes had lowest annual levels in the warmer months in August–September (Figure 5).

#### 2.3.3. PCA and Correlative Responses of Seagrass Traits

Overall, the PCA analysis based on morphological, leaf productivity, population, and biochemical descriptors revealed clear differences between ZM1 and ZM2, resulting in distinct separation of the two ecotypes and seasonally collected samples (Figure 6).

The correlation matrix, based on Pearson’s coefficient, allowed us to examine the variables that were more closely correlated based on the combined data from both ecotypes. (Figure 7). Additionally, the correlation matrix for each ecotype is provided in the Appendix A. Particularly, there was a positive and significant correlation between NSC and leaf biomass production (Deming regression, *n* = 11, R2 = 0.362, *p* < 0.05) (Figure 8A). Also, there was a negative and significant correlation between omega-3 and leaf production (Deming regression, *n* = 11, R2 = 0.642, *p* < 0.05) (Figure 8B). Similarly, a negative and significant correlation was observed between TFA and TCH (Deming regression, *n* = 11, R2 = 0.708, *p* < 0.05) (Figure 8C). Additionally, there was a positive correlation between SFA and TCH (Deming regression, *n* = 11, R2 = 0.469, *p* < 0.05) (Figure 8D). Lastly, we observed two negative and significant correlations between the unsaturation index 18:3 *n*-3/16:0 and LAI and NSC (Figure 8E,F).

## 3. Discussion

This study characterized the relationship between the seasonality of leaf biomass production and synthesis of non-structural carbohydrates (NSC) and fatty acids (FAs) in two co-existing ecotypes of *Zostera marina*. Previous studies have investigated seasonal changes in individual biochemical compounds in seagrasses, including carbohydrates and fatty acids (e.g., [23,26,29,37]); however, possible correlations between these functional components have not previously described. Our findings indicate that their synthesis is driven by the interaction of climatic factors, vegetative development, and differential ecological strategies of the two ecotypes.

### 3.1. Seasonal Responses of Seagrass Traits to Environmental Drivers

Seasonal changes related to temperature and irradiance were identified as the main factors modulating the seasonal responses of seagrass traits. Such results could be expected, as seagrasses in temperate regions are known to adapt their morphology, physiology, and biochemical composition in response to environmental conditions throughout the year [23,28,29,55].

Temperate seagrasses undergo significant reductions in leaf size and total area during unfavorable winter conditions which are typically characterized by annual low sea surface temperature and irradiance. Additionally, marked reductions in NSC and SFAs contents are typically observed during colder and darker periods, which enable plants to maintain low metabolic demands and support respiration [28,37,56]. The utilization of stored energetic compounds to survive cold and unfavorable growth conditions has previously been reported in *P. oceanica* and *Z. noltei* during an annual cycle [57,58,59]. Seagrasses store NSC in both aboveground tissues (leaves) and belowground tissues (roots and rhizomes), and it is expected that potential remobilization between these tissues may occur in response to physiological demands throughout the annual cycle [60,61]. In parallel, during winter, both ecotypes experienced a 77% increase in average TFA content compared to summer, which is consistent with previous findings [37]. Such an increase in TFAs was primarily explained by an increase in PUFAs relative to SFAs; particularly, *n*-3 PUFAs increased by 37.5% from their minimum values in August–September to their maximum in January. The reduction in temperature and irradiance levels triggered an increase in unsaturation levels within seagrass leaves, facilitating optimal adaptation to less favorable environmental conditions [62,63,64,65].

In spring, both ecotypes exhibited the highest TFA and *n*-3/*n*-6 PUFA values. This season is characterized by favorable environmental conditions for seagrass growth and productivity due to high nutrient concentrations and the formation of the thermocline [66,67,68,69]. Seagrasses adapt their growth and biochemical composition according to environmental conditions and precondition their seasonal performance to internal biological rhythms [70]. Thus, these seasonal patterns represent an adaptive mechanism for seagrasses to optimize the use of resources in the water column and to physiologically prepare for optimal environmental conditions in warmer months [41,71].

During warmer conditions coinciding with the highest irradiances in summer, both *Z. marina* ecotypes grew more and also accumulated higher amounts of NSC and SFAs, as was previously observed in Mediterranean seagrasses [23,72,73]. Generally, larger leaf structures developed under optimal environmental conditions, which maximized photosynthetic activity and resulted in excess energy generated being transformed into energetic compounds such as NSC or SFAs (i.e., [28,41,71,74]). Interestingly, for both ecotypes, NSC decreased briefly in early summer, which appeared to be associated with a sharp reduction in temperature and irradiance as was previously reported in Danish *Z. marina* [75], suggesting a fast and direct response to ambient conditions.

We observed a significant decrease in unsaturation levels in leaves during warmer months, primarily associated with increases in SFAs relative to PUFAs and reductions in *n*-3 PUFA relative to *n*-6 PUFA compared to colder and darker periods. Generally, marine and terrestrial primary producers reduce unsaturation requirements to achieve optimal membrane fluidity and photosynthetic performance at higher temperatures [33,35,76,77]. In seagrasses, the accumulation of high levels of SFAs at higher temperatures also appears to aid the production of PUFAs under suboptimal thermal environmental conditions, as SFAs are partially the precursors of PUFAs [41].

Recent studies have demonstrated the usefulness of fatty acids as a reliable ecological indicator of seagrasses exposed to temperature, irradiance, acidification, and nutrient stress (e.g., [41,78,79,80,81,82].) Irish *Z. marina* plants experienced a maximum summer SST of 19 °C, which is within the temperate range of this species [19], suggesting that our results do not likely reflect critical physiological stress but can be used as a proxy to evaluate seasonal changes in the eco-physiological state of this seagrass species.

### 3.2. Correlative Responses of Seagrass Traits

To our knowledge, this is the first study to investigate potential seasonal correlations between energetic compounds in seagrasses. Seasonal analysis of *Z. marina* including the data of both ecotypes revealed a negative correlation between NSC and TFA, which can be largely attributed to variations in PUFA and *n*-3 PUFA levels. This pattern is likely due to the fact that *Z. marina* has higher levels of unsaturated FA in their leaves in response to colder conditions, while also utilizing energy reserves such as NSC to survive less favorable environmental conditions [28,37]. Noteworthy, the synthesis of PUFAs typically involves a series of desaturation and elongation reactions that convert shorter, saturated, or monounsaturated fatty acids into longer, more highly unsaturated fatty acids, which is a process that requires significant energy consumption [41,74]. Conversely, a positive seasonal correlation was observed between the production of NSC and SFAs, indicating that the accumulation and utilization of these energetic compounds followed the same pattern throughout the year [83,84]. The positive relationship between SFAs and NSC can be attributed to their roles in energy storage and utilization, with SFAs serving as a secondary energy source when NSC levels are low [85].

In addition, we observed a negative seasonal correlation between leaf biomass production and the accumulation of *n*-3 PUFAs. Notably, non-structural carbohydrate content, specifically starch, showed a negative correlation with TFA levels in leaves. This pattern suggests that part of the energetic compounds accumulated under favorable growth conditions is invested in the production of essential PUFAs to cope with sub-optimal winter conditions. However, further research is needed to validate this hypothesis.

### 3.3. Ecotype-Specific Responses of Seagrass Traits

Our findings validate the perennial ecological strategy of ecotype ZM1 and the annual pattern exhibited by ecotype ZM2, thus confirming the latter as the *Zostera marina* angustifolia ecotype. Despite the close proximity of the meadows (~20 m), they are physically separated, each inhabiting a distinct area.

The two ecotypes of *Z. marina* exhibited significant differences in the seasonal responses of leaf area, leaf biomass production, and leaf area index (LAI) throughout the year. Such variations may be attributed to their distinct ecological strategies; for instance, ZM2, the smaller ecotype with thinner leaves, typically inhabits the lower intertidal in Ireland, while the common ecotype of *Z. marina* colonizes subtidal areas. It has previously been suggested that morphological adaptations of seagrasses can occur in response to air exposure where, for example, production of smaller leaves in intertidal zones appeared to prevent damage from desiccation stress [25,86]. In this study, ZM1 exhibited higher values of leaf area, biomass production of photosynthetic tissues, and LAI than ZM2 throughout the year. ZM1 was able to progressively regrow from a minimum in February to a maximum in summer; by contrast, ZM2 showed delayed growth until May, after reaching its minimum in January. Furthermore, ZM1 maintained maximum seasonal levels of LAI from July to September and changed more gradually, while LAI of ZM2 increased sharply from a minimum to a maximum in September, and then declined rapidly. These observations suggest that ZM1 had a more consistent growth pattern than ZM2 throughout the year. In winter, sucrose and starch contents of ZM2 decreased drastically, containing about 11 times less sucrose than in August. By contrast, ZM1 was able to retain significantly larger amounts of NSC in photosynthetic tissues during the colder months, indicating more stable seasonal variation of these vital compounds.

Overall, the FA profiles and unsaturation patterns of the two studied ecotypes of *Z. marina* were similar, but interestingly with a contrasting pattern of the unsaturation index (18:3 *n*-3/16:0), which is likely related to their different vegetative strategies. ZM1 exhibited the highest level of unsaturation based on the unsaturation index (18:3 *n*-3/16:0) in early spring, at the start of its growing season, while ZM2 had its highest level of unsaturation in early summer, at the beginning of its growth phase. These results suggest that this index was linked to the growth phase rather than to thermal regimes, confirming the reliability of this indicator as an eco-physiological indicator of seagrasses, as previously described by [81]. According to these findings, ecotype-specific trends suggest that changes in unsaturation levels in photosynthetic structures are likely related to (i) the optimization of environmental resources to favor vegetative development of the plant, and (ii) physicochemical adjustments of photosynthetic structures to cope with less favorable environmental conditions. This finding is particularly novel and relevant as it deviates from the conventional understanding of FA indicators in seagrasses, which were typically associated with environmental changes such as temperature, irradiance, or nutrient concentration. However, this discovery highlights a new perspective, indicating that ecological strategies can also play a role in influencing adjustments to unsaturation levels in the photosynthetic structures of seagrasses.

Developing ecological indicators related to plant traits, such as NSC and FAs, is critical in times of global change, as these indicators can provide valuable insights into the impacts of environmental changes on marine and terrestrial plants [80,87]. Changes in target FAs and alterations in NSC allocation of plant tissues can reflect changes in environmental conditions, energy resources, and plant stress responses. Therefore, by monitoring these ecological indicators, researchers can better elucidate the impacts of global change on plant health and functioning.

## 4. Materials and Methods

### 4.1. Study Area

The study was conducted in Kilkieran Bay in the Connemara National Park, northern Galway Bay (53°19′35″ N; 9°36′58″ W), a site that is designated as a Special Area of Conservation (SAC) and Natural Heritage Area (NHA) due to the presence of seagrass meadows (Table 4) and other ecologically important marine and terrestrial ecosystems. In Connemara, approximately 18 km^2^ of seagrasses have been documented [47]. The majority of these meadows consist of subtidal *Z. marina* meadows, with no recorded presence of *Z. noltei*. Notably, there have been no descriptions of intertidal meadows in this region to date.

Specifically, the study site is semi-exposed, with tidal ranges of 4–5 m, and the seagrass meadows were located in the intertidal area in muddy and sandy sediments in a sheltered location. In the study site (Figure 9), seagrass meadows extend across an area dominated by rock formations and small cliffs ranging from 2 to 5 m in height. Within this coastal-scape, there are scattered sheltered zones, typically small areas measuring around 20 to 100 m^2^, characterized by muddy and sandy sediments, occasionally colonized by seagrasses. The study specifically targeted two distinct meadows, each colonized by two different ecotypes of *Z. marina*, referred to as ZM1 and ZM2. ZM1 is the common ecotype of *Z. marina*, characterized by persistent leaf structures throughout the year (i.e., perennial population), and forms a meadow with an area of 2900 m^2^. ZM2 is the ecotype previously known as *Z. marina* var. angustifolia, located in a smaller meadow with an area of 1700 m^2^. ZM2 displays an annual growth pattern, with almost complete loss of photosynthetic structures during winter (based on personal observation). Both ecotypes are exposed to similar environmental conditions, including depth, sediment type, and hydrodynamics. The meadows of the two ecotypes are physically separated by rocks, located in two distinct sheltered zones, with a distance of approximately 20 m between them (Figure 9).

### 4.2. Environmental Variables

The western Irish climate is defined as temperate and is characterized by mean sea temperatures ranging from approximately 6 to 20 °C, with a lack of extreme warm or cold seasons. Data of daily sea surface temperatures (SST) were obtained from the National Oceanic and Atmospheric Administration (https://coastwatch.pfeg.noaa.gov/erddap/griddap/jplMURSST41.html; accessed 15 May 2020), and those of daily global irradiation on the horizontal plane at ground level (GHI) by Copernicus Atmosphere Monitoring Service (CAMS radiation service—SoDa (soda-pro.com; accessed 4 June 2020)).

### 4.3. Sampling Procedure

Sampling was carried out monthly between November 2017 and October 2018 using both on-foot and snorkeling approaches. To assess morphometric and productivity descriptors (details below), we randomly marked ~15 healthy apical shoots along each transect and collected them every month using the punching method. To facilitate identification, we attached a plastic tie to the base of each punched shoot. For biochemical analyses (details below), we monthly collected 10 random shoots along each permanent transect.

Samples were then transported to the laboratory on the University of Galway campus (1 h drive), in zip-lock bags filled with cooled seawater to protect leaf materials. In the laboratory, samples were kept in seawater containers in constant temperature rooms set to temperatures similar to the in situ conditions. Prior to processing, samples were washed with filtered seawater to remove any epiphytes or sediment.

#### 4.3.1. Morphometric Descriptors

To calculate the total leaf area (cm^2^ shoot^−1^), we summed the area of all individual leaves belonging to the same shoot, which was measured by multiplying the length and width of each leaf. For ZM1, length and width data were extracted from [55] (Appendix A). Specifically, we measured the width at the midpoint of the length of the second youngest leaf.

#### 4.3.2. Productivity Descriptors

For in situ leaf biomass production, we used the punching protocol (e.g., [55]) which involved making two small holes in the stem above the basal meristem of the plant with tweezers. We then weighed the total leaf material (dry weight [DW]) produced in situ over a period of 28–31 days and normalized the results per day using the following equation (Equation (1)):(1)$$Leaf\;production\;(LP)=\frac{\sum leaf\;biomass}{t_{f}-t_{0}}$$

LP is leaf production (mg DW day^−1^ shoot^−1^); leaf biomass corresponds to the newly produced and weighed dry leaf biomass after 48 h in an oven (60 °C); *t*_f_ − *t*_0_ refers to the marked growth period in days.

#### 4.3.3. Population Descriptors

We calculated the leaf area index (LAI) by multiplying the total leaf area per shoot (m^2^) by the shoot density within the seagrass meadow (shoots m^−2^). To determine shoot density, we counted the total number of shoots in a randomly placed quadrat (0.33 m^2^) 3–5 times along the transect at each monthly sampling event. For ZM1, shoot density data were extracted from [55] (Appendix A).

### 4.4. Biochemical Analyses

Prior to conducting biochemical analyses, material from the second youngest leaf of each shoot (*n* = 3–4) were collected and cleaned with filtered seawater. For biochemical analyses, plants were randomly collected in four different locations along the permanent transect. The selected leaves were then dried with a paper towel, frozen at −20 °C for 48 h, and finally freeze-dried for 24–48 h using a Labconco Freezone Freeze-dryer System (Kansas City, MO, USA). The freeze-dried samples were then kept at −40 °C until further use.

#### 4.4.1. Fatty Acid Analyses

To analyze the fatty acid (FA) content and composition, we used a protocol previously applied to macroalgae and seagrasses [88,89,90] based on the analysis of fatty acid methyl esters (FAMEs) by direct transmethylation. Before the initial extraction process, freeze-dried biomass was powdered using a Beadmill 4—Fisher Scientific (Hampton, NH, USA) machine (five ms^−1^ for 2 min). Then, we carried out extractions with 2 mL of dry methanol, containing 2% (*v*/*v*) H_2_SO_4_, which we carefully pipetted into each separate vial containing 25 mg (±10 mg) of dry biomass. Then, 10 μL of the FA C15 (pentadecanoic acid) was added to each transmethylation vial as an internal standard. To prevent FA oxidation and degradation, vials were closed under nitrogen atmosphere, and vials containing the samples were heated at 80 °C for 1.5–2 h. After transmethylation, we extracted the FAME which was analyzed using an Agilent 7890A GC/5975C (Santa Clara, CA, USA) mass selective detector (MSD). We identified the FAME using co-chromatography with authentic commercially available FAME standards (Supelco™ 37 Component FAME Mix, catalogue no. 47885-U, Supelco, Bellefonte, PA, USA) and FAME of fish oil (Menhaden Oil, catalogue no. 47116, Supelco).

#### 4.4.2. Non-Structural Carbohydrates Analyses

Non-carbohydrate content (NSC) was determined following [29]. First, we extracted free sugars (sucrose) in 80% boiling ethanol, and then evaporated the extracts at room temperature. Extracts were then redissolved in distilled water and analyzed in a spectrophotometer (UNICAM UV-1700 Pharma Spec, Shimadzu Corporation, Kyoto, Japan) using a sucrose resorcinol standard assay [91]. To obtain starch, we extracted the ethanol-insoluble fraction overnight in 1 N NaOH and analyzed it with a spectrophotometer (UNICAM UV-1700 Pharma Spec) using an anthrone assay that was standardized for sucrose [92].

### 4.5. Statistical Analysis

Before comparing seasonal responses of selected seagrass traits (morphological, production, and biochemical compounds of the two *Zostera marina* ecotypes), we first assessed the homogeneity of variances using Levene’s test and normality using the Shapiro–Wilk test. As data did not meet the criteria for normality and homogeneity of variance, a non-parametric PERMANOVA test was used based on a similarity matrix with Euclidean distances. We implemented a two-way factorial design with two fixed factors: ‘month’ (M) and ‘ecotype’ (E), and performed a pairwise test to compare monthly responses.

A potential correlation between the produced photosynthetic tissues and the biochemical composition of this newly produced leaf material was explored by calculating a correlation matrix based on Pearson’s correlation coefficient.

To investigate whether SST and irradiance explained seasonal responses of each ecotype (ZM1 and ZM2) of the selected seagrass traits, including morphology, population descriptors, biomass production, carbohydrates, and fatty acids content and composition, we implemented general linear models (GLMs). We assumed a Gaussian distribution because we used negative values in the set of dependent variables.

Potential differences or similarities between ecotypes (ZM1 and ZM2) regarding seasonal patterns of the studied descriptors were visualized after conducting principal component analyses (PCA) based on Euclidean similarity as a flexible ordination technique. Prior to PCA analyses, data were normalized to ensure equal contributions of each seagrass descriptor.

All data treatments and statistical analyses were conducted using R software (R Core Team, https://cran.r-project.org/ (assessed on 11 May 2020)) [93] and PRIMER and PERMANOVA 6 [94].

## 5. Conclusions

This study provides valuable insights into the complex relationship between non-structural carbohydrate content, fatty acid profiles, and plant production in seagrass tissues, and seasonal compound patterns were specific to ecotype. The positive correlation between total non-structural carbohydrates and saturated fatty acids, both of which serve as energetic components, suggests that they are simultaneously involved in energy storage and utilization. Monitoring the eco-physiological and health status of seagrass populations requires an understanding of the seasonal dynamics and biochemical adaptations that occur in response to changing environmental conditions. Our findings underscore the importance of developing novel ecological indicators to assess the impacts of global change on seagrass ecosystems.

## Figures and Tables

**Figure 1 plants-13-00396-f001:**
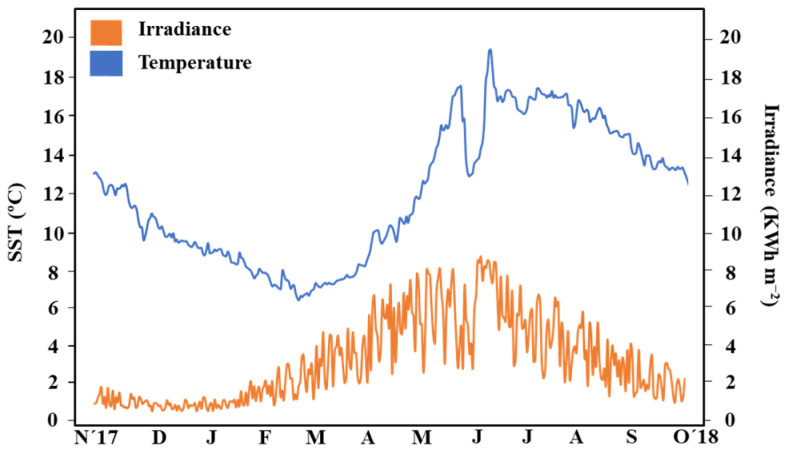
Daily average of sea surface temperature (SST (°C)) and daily irradiance (kWh m^−2^) in Kilkieran Bay (north Galway Bay), Ireland (53°19′35″ N; 9°36′58″ W) from November 2017 to October 2018.

**Figure 2 plants-13-00396-f002:**
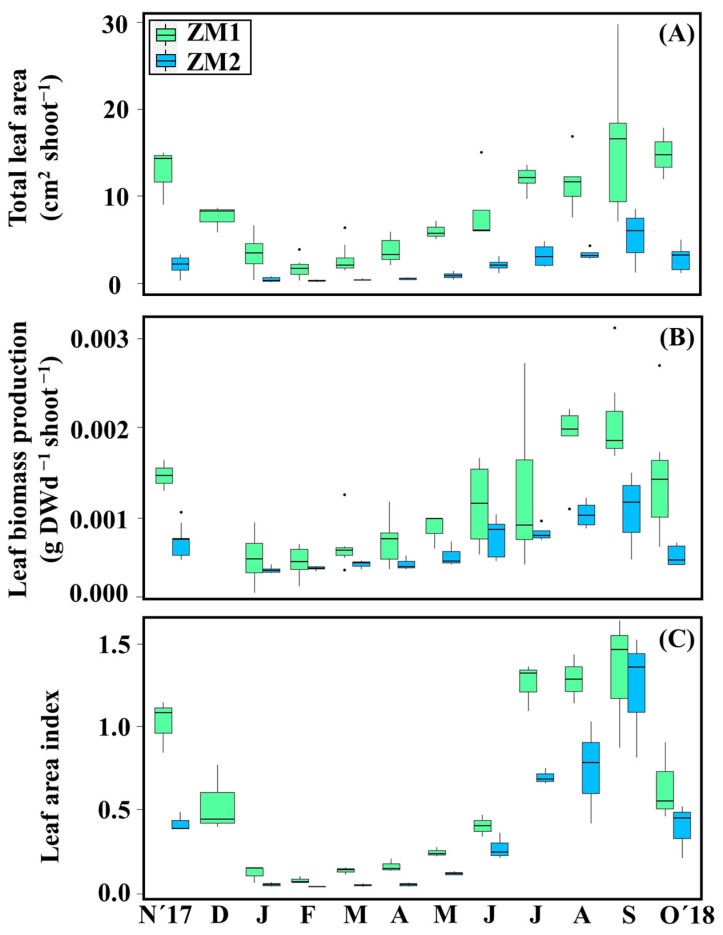
Trends of total leaf area (cm^2^ shoot^−1^) (**A**). Leaf biomass production (g DW d^−1^ shoot^−1^) (**B**). Leaf area index (m^2^ m^−2^) (**C**). Results are expressed as mean ± SD (*n* = 3–13). Black points are considered outliers.

**Figure 3 plants-13-00396-f003:**
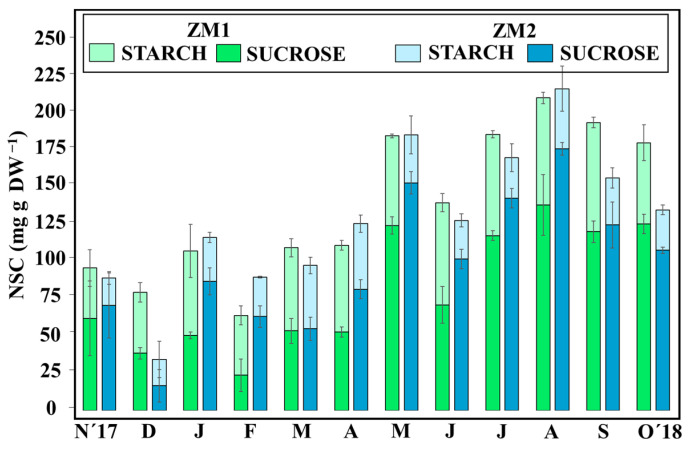
Total non-structural carbohydrate content (sucrose and starch) in leaves. Results are expressed as mean ± SD (*n* = 3).

**Figure 4 plants-13-00396-f004:**
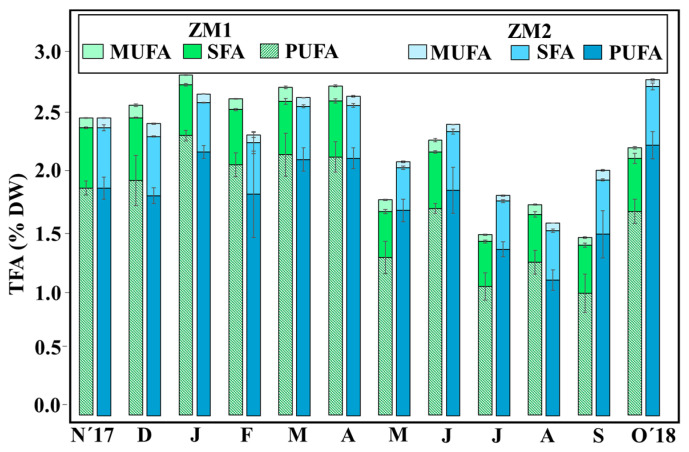
Total fatty acids (TFA) content (%DW) and composition (%TFA) in leaves (MUFA, SFA, and PUFA). Results are expressed as mean ± SD (*n* = 3–4).

**Figure 5 plants-13-00396-f005:**
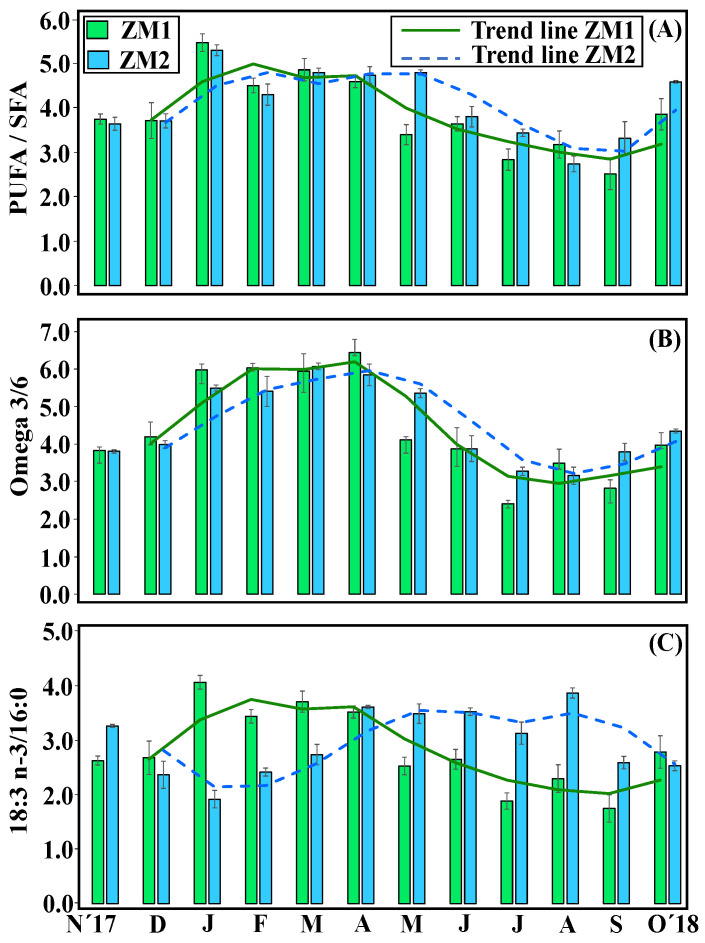
Seasonal trends in leaves; polyunsaturated FA/saturated FA (PUFA/SFA) ratios (**A**). Omega-3/6 ratios (**B**). 18:3 *n*-3/16:0 FA ratios (**C**). Results are expressed as mean ± SD (*n* = 3–4).

**Figure 6 plants-13-00396-f006:**
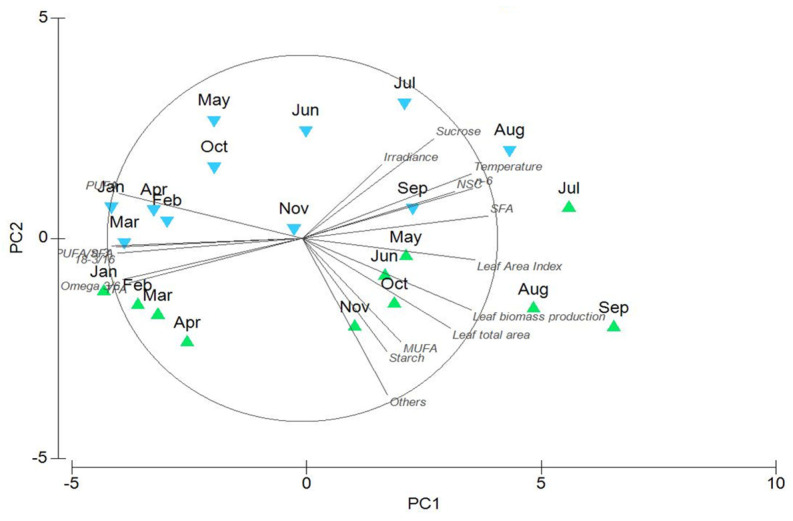
Results of PCA displaying differences and similarities across seasonal samples of ecotype 1 (ZM1) and ecotype 2 (ZM2) for all seagrass descriptors (total leaf area, leaf area index, leaf biomass production, sucrose, starch, TFA, PUFA/SFA, and omega-3/6).

**Figure 7 plants-13-00396-f007:**
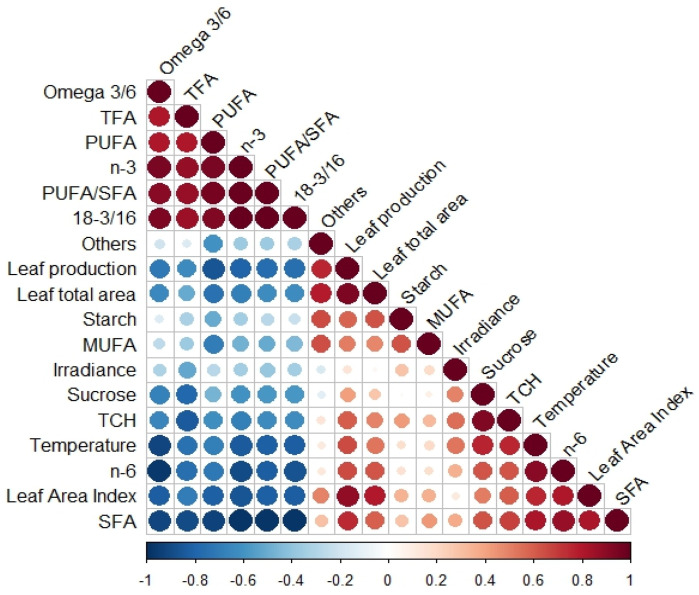
Correlation matrix based on Pearson’s coefficient correlating seagrass descriptors, irradiance, and SST. Positive correlations are represented in red and negative in blue. The higher size of the circles indicates a higher correlation between the compared variables (*n* = 24).

**Figure 8 plants-13-00396-f008:**
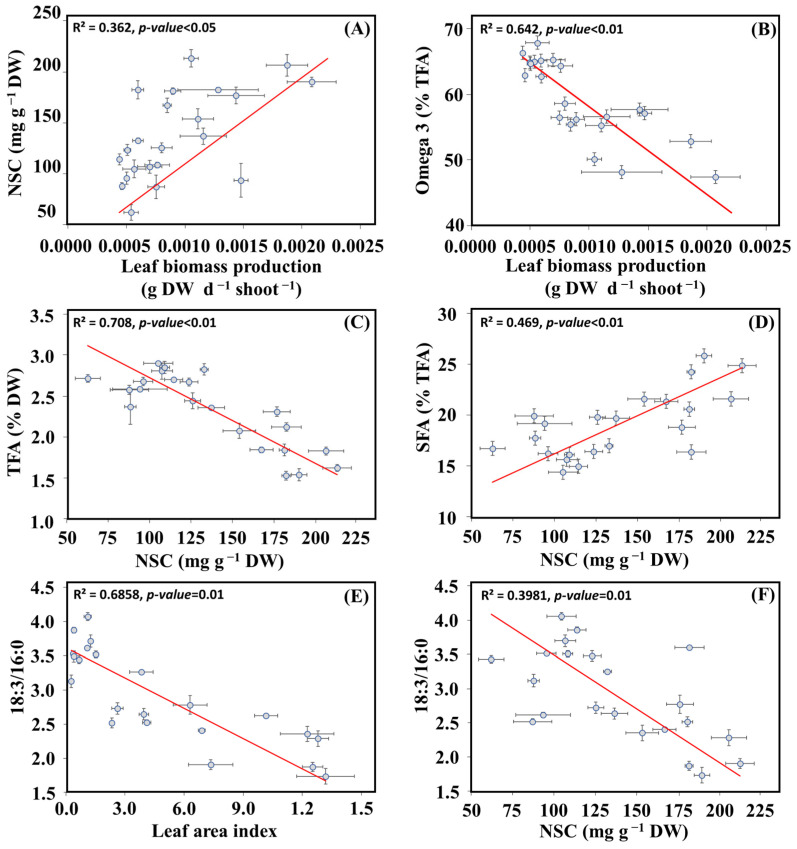
Deming correlation of non-structural carbohydrates (NSC) versus leaf biomass production (**A**). Omega-3 (*n*-3 PUFA) versus leaf biomass production (**B**). Total fatty acids (TFA) versus non-structural carbohydrates (NSC) (**C**). Saturated fatty acids (SFAs) versus non-structural carbohydrates (NSC) (**D**). 18:3/16:0 ratio versus leaf area index (**E**). 18:3/16:0 ratio versus non-structural carbohydrates (NSC) (**F**). Red lines represent the Deming regression line (*n* = 24).

**Figure 9 plants-13-00396-f009:**
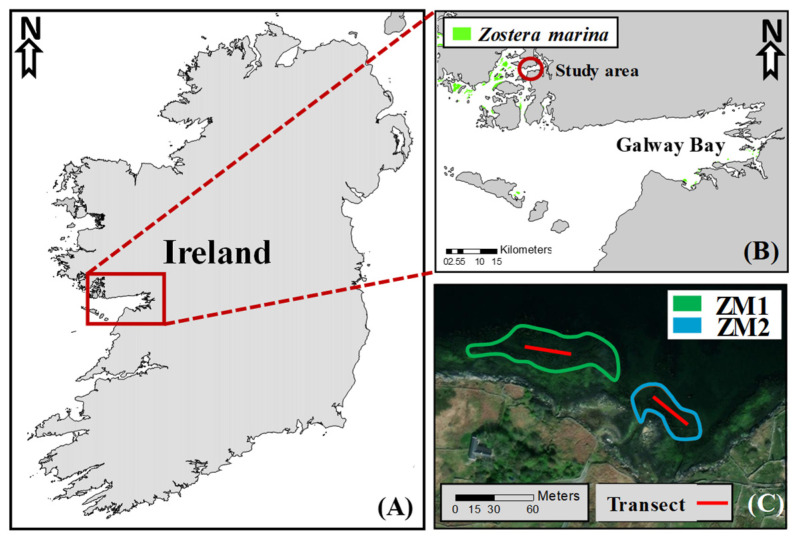
Map of Ireland (**A**). The red rectangle represents the study area (53°19′35″ N; 9°36′58″ W) in Kilkieran Bay, Co. Galway, showing the general distribution of seagrass meadows (green) (**B**). The ecotype 1 (ZM1) is represented in green, and the ecotype 2 (ZM2) in blue (**C**).

**Table 1 plants-13-00396-t001:** Effect of factors ‘month’ (M), ‘ecotype’ (E), and their interaction in morphometric and productivity descriptors: total leaf area, leaf biomass production, and leaf area index of ecotype 1 (ZM1) and ecotype 2 (ZM2). Pseudo-F values of two-way PERMANOVA are shown along with significance levels (*** *p* < 0.001).

	Total Leaf Area (cm^2^ shoot^−1^)			Leaf Production (g DW d^−1^ shoot^−1^)			Leaf Area Index (m^2^ m^−2^)		
**Treatment**	df	MS	Pseudo-F	df	MS	Pseudo-F	df	MS	Pseudo-F
Month (M)	11	4.7	22.6 ***	10	6.1	16.4 ***	11	4.9	40.6 ***
Ecotype (E)	1	40.4	194.8 ***	1	19.9	53.2 ***	1	4.3	35.3 ***
MxE	10	1.5	7.3 ***	10	1.1	3.0 ***	10	0.3	2.7 ***
Residual	117	0.2		114	0.4		46	0.1	
Total	139			135			68		

**Table 2 plants-13-00396-t002:** Effect of ‘month’ (M), ‘ecotype’ (E), and their interaction on non-structural carbohydrates (sucrose and starch) of ecotype 1 (ZM1) and ecotype 2 (ZM2). Pseudo-F values of two-way PERMANOVA are shown along with significance levels (* *p* < 0.05; *** *p* < 0.001).

		Sucrose (mg g^−1^ DW)		Starch (mg g^−1^ DW)	
**Treatment**	df	MS	Pseudo-F	MS	Pseudo-F
Month (M)	11	5.5	64.2 ***	1.6	6.5 ***
Ecotype (E)	1	2.4	28.7 ***	35.4	140.9 ***
MxE	11	0.4	4.7 ***	0.5	2.0 *
Residual	48	8.5 × 10^−2^		0.2	
Total	71				

**Table 3 plants-13-00396-t003:** Effect of ‘month’ (M), ‘ecotype’ (E), and their interaction on total fatty acids (TFAs) content and FA composition (PUFA, MUFA, SFA, PUFA/SFA, omega-3, omega-6, omega-3/6, 18:3 *n*-3/16:0) on leaves ecotype 1 (ZM1) and ecotype 2 (ZM2). Pseudo-F values of two-way PERMANOVA are shown along with significance levels (** *p* < 0.01, *** *p* < 0.001).

		TFA		PUFA		MUFA		SFA			
**Treatment**	df	MS	Pseudo-F	MS	Pseudo-F	MS	Pseudo-F	MS	Pseudo-F		
Month (M)	11	6.2	37.9 ***	5.3	36.6 ***	2.4	14.5 ***	6.5	55.8 ***		
Ecotype (E)	1	0.1	0.9	13.0	90.4 ***	24.4	149.1 ***	0.6	5.3 **		
MxE	11	0.8	4.8 ***	0.9	6.1 ***	2.7	16.6 ***	0.9	8.0 ***		
Residual	69	0.2				0.2		0.1			
Total	92										
		**PUFA/SFA**		**Omega-3**		**Omega-6**		**Omega3/6**		**18:3 *n*-3/16:0**	
Treatment	df	MS	Pseudo-F	MS	Pseudo-F	MS	Pseudo-F	MS	Pseudo-F	MS	Pseudo-F
Month (M)	11	6.7	68.5 ***	6.5	65.9 ***	7.3	100.4 ***	7.3	110.2 ***	6.8	74.2 ***
Ecotype (E)	1	1.9	19.1 ***	2.3	23.2 ***	0.3	3.7	0.2	2.8	0.8	8.9 ***
MxE	11	0.8	7.8 ***	0.9	9.2 ***	0.5	7.4 ***	0.5	7.8 ***	0.8	9.0 ***
Residual	69	9.8 × 10^−2^		9.9 × 10^−2^		7.3 × 10^−2^		6.6 × 10^−2^		9.2 × 10^−2^	
Total	92										

**Table 4 plants-13-00396-t004:** Details of location, sampling period, coordinates, meadow area (ha), and minimum and maximum sea surface temperature (SST [°C]) and irradiance (Wh m^−2^) of the study site and the studied ecotype 1 (ZM1) and the ecotype 2 (ZM2).

Country	Location	Sampling		Coordinates		Area		SST		Irradiance	
		Period				(m^2^)		(°C)		(Wh m^−2^)	
						ZM1	ZM2	Min.	Max.	Min.	Max.
Ireland	Kilkieran	Nov	Oct	53°19′35″ N	9°36′58″ W	2900	1700	6.8 ± 0.4	16.8 ± 0.4	473.6 ± 173.1	5823.9 ± 1920.1
	Bay (Galway)	2017	2018								

## Data Availability

Data are contained within the article and Appendix A.

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
