# Peer review of "Ecotype-Specific and Correlated Seasonal Responses of Biomass Production, Non-Structural Carbohydrates, and Fatty Acids in Zostera marina"

_plants, 2024, doi:10.3390/plants13030396_

Round 1

Reviewer 1 Report

Comments and Suggestions for Authors

The research represents a contribution to the knowledge of the eco-physiological responses to seasonal environmental variation and its correlation with biochemical and biomass production traits in two different ecotypes of the eelgrass Zostera marina.

The manuscript is well organized and written except for minor typing errors that are reported in the attached file. There are some minor methodological concerns that should be addressed by the authors and the consistency in numbering of figures and tables should be checked to reflect the order in the manuscript and the supplementary files. Also, the use of italics for scientific names should be checked.

A weakness of the study would be the lack of further evidence (such as genetic data) that supports the claim of ecotypic differentiation. This could be mentioned as a limitation of the study. However, I understand that would be the subject of a different study and, if there is already one for the same meadows, it should be cited to improve this manuscript.

Comments on the Quality of English Language

The use of the language is appropiate requiring just a minor editing.

Author Response

Reviewer 1

Comment 1: A weakness of the study would be the lack of further evidence (such as genetic data) that supports the claim of ecotypic differentiation. This could be mentioned as a limitation of the study. However, I understand that would be the subject of a different study and, if there is already one for the same meadows, it should be cited to improve this manuscript.

Answer: We appreciate your concern. In fact, we conducted genetic analyses to confirm that both ecotypes were indeed Z. marina. Please refer to the provided analyses, which demonstrate slight differences between the two ecotypes, but in both cases, the results were very similar. Please see the analyses below:

“DNA extraction, amplification, sequencing, and identification of five Zostera were received at the Molecular Biology peripheral service of INMAR on 07/04/2021.

DNA extraction from the samples was carried out following the instructions of the DNeasy Plant Mini Kit from the commercial brand Qiagen, resulting in a final volume of 200 μl. Partial amplification of the rbcL gene using the polymerase chain reaction, or PCR, was performed in a volume of 25 μl, containing: 0.25 μl of Taq (5u/μl) from the commercial brand Qiagen, 2.5 μl of CoralLoad PCR Buffer, 2.5 μl of dNTPs (2mM), 5 μl of Q-solution, 3.5 μl MgCl (25mM), 1 μl of each primer (10uM), 2 μl of the sample, and 7.25 μl of Milli-Q water. The primers used for each of the samples are specified below:

Zoostera rbcL-F (ATGTCACCACAAACAGAGACTAAAGC) y rbcL-R

(GCAGCAGCTAGTTCCGGGCTCC)

Table 1: Information on sequencing results and BLASTn in GenBank:

Zoostera 233 (ZM1) ✓ Z. marina, Z. angustifolia, Z. caulescens, Z. asiática, Z. caespitosa, Z.

japonica (100%)

Zoostera 234 (ZM1) Z. marina, Z. angustifolia, Z. caulescens, Z. asiática, Z. caespitosa, Z.

japonica (100%)

Zoostera 235 (ZM2) ✓ Z. marina, Z. angustifolia, Z. caulescens, Z. asiática, Z. caespitosa

(above del 99%)

Zoostera 236 ✓ (ZM2) Z. marina, Z. angustifolia, Z. caulescens, Z. asiática, Z. caespitosa

(above 99%)

Zoostera 237 ✓ (ZM2) Z. marina, Z. angustifolia, Z. caulescens, Z. asiática, Z. caespitosa

(above 99%)

Specific Comments

Specific Comments Line 31-32: did the authors mean “unsaturation levels of fatty acids in seagrass leaves”?

Yes, thanks, we have change it.

Line 111: Scientific name in italics ,

We hare change in all the article.

Line 127: Journal will have the final saying in formatting, but it would be useful for the reader if tables and figures are numbered in order of appearance.

We have reordered the tables and figures in the article to match their appearance in sequence.

Line 136: Abbreviations ZM1, ZM2, LAI have not been defined before. They should be defined here as it is the first time the appear in the main manuscript.

We agree, we have changed.

Line 167: Does “(n=3-13)” means a different sample size in different months?

Yes, typically, we collected around 10 samples per transect. However, during some winter months with extreme storms, a few marks or plants disappeared. The occurrence of only 3 samples was only observed in the month of January.

Line 192: In the main text Table S3 refers to ‘total fatty acids’ but in the supplementary files it corresponds to sucrose and starch. This should be corrected to Table S4.

Changed, thanks.

Line 209: The caption states that the units are “%DW” but the figure shows “mg g-1DW” this should be clarified.

Thanks, it should be %DW.

Line 285-287: This should be rephrased as it is a speculative assumption, no data regarding the thylakoid and the photosynthetic activity was object of this study.

We agree, we have rephrased the sentence “The reduction in temperature and irradiance levels triggered an increase in unsaturation levels within seagrass leaves, facilitating optimal adaptation to less favourable environmental conditions.”

Line 320: add: “this” is the first study…

Thanks, added

Line 437: this is confusing. Was the area (length and width) of all individual leaves belonging to the same shoot measured or just the second youngest leaf of each shoot?

We measured the length of each individual leaf of the plant, while the width was exclusively measured in the second leaf, with this measurement later applied to all the leaves. Typically, the width of Zostera marina remains relatively constant along the leaves, making it a suitable measurement for this purpose.

Line 452: isn’t “leaf area index” a dimensionless ratio (m2 / m2)?

Yes, you are correct, but in some papers, it appears with units, and in others, it does not. We have eliminated it from the article.

Line 464: this phrase is repetitive, should be better to delete it.

Deleted, thanks

Line 470: freeze-dried biomass?

Yes, thanks, added.

Section 4.3. It would be useful to describe how many transects per meadow were used and if possible, to add the location of transects in figure 1.c.

We employed one transect per meadow due to the small size of the meadows. The figure now includes the location where we positioned the permanent transect.

Reviewer 2 Report

Comments and Suggestions for Authors

In this study, the authors analyze two Zostera marina ecotypes cohabiting in a seagrass meadow in northern Galway Bay, Ireland. The two ecotypes were seasonally characterized based on shoot leaf area, biomass production, leaf area index, and content of biochemical compounds (carbohydrates and FAs).

This is a very interesting study and is very well written. Since I believe most of my comments can be addressed through appropriate discussion, I think this manuscript will be acceptable after minor revision.

How did you define the two Zostera marina ecotypes, and given their proximity, how do you explain their cohabitation? Please address these questions in the introduction and discussion sections.

Minor comments are provided in the attached PDF.

Author Response

Reviewer 2

How did you define the two Zostera marina ecotypes, and given their proximity, how do you explain their cohabitation? Please address these questions in the introduction and discussion sections

Thank you for the suggestion. We have incorporated new information into the introduction, discussion, and methods.

In lines 93 – 100 we have rewrite the paragraph adding the required information.

In lines 141-144 we added: “In this study, we characterized two Z. marina ecotypes: one representing the perennial Z. marina ecotype, identified as ZM1, and the second ecotype, representing the annual and smaller Z. marina ecotype known as Z. marina angustifolia, designated as ZM2 (further details are outlined in the section 4).”

In lines 360-363 we added “Our findings validate the perennial ecological strategy of ecotype ZM1 and the annual pattern exhibited by ecotype ZM2, thus confirming the latter as the Zostera ma-rina angustifolia ecotype. Despite the close proximity of the meadows (~20m), they are physically separated, each inhabiting a distinct area.”

To clarify: The two seagrass meadows, one colonised by the ecotype ZM1 and other by the ecotype ZM2, are physically separated by rocks and situated in two different small “bays” by a distance of 20m approximately.  Therefore, they are not cohabitating. We monitored these meadows for three years before initiating the proposed study. Based on this experience, we were able to identify that they represent two different ecotypes. In the study area, and generally in western Ireland, the predominant ecotype is the perennial Z. marina (like ZM1), being the presence of Z. marina Angustifolia quite unusual. Indeed, this meadow is the only one documented in a large area with these characteristics. Initially, we thought the annual meadow was Z. noltei, but we performed genetic analyses to confirm that it was Z. marina. Please find the attached response to reviewer 1. Due to this singularity, we decided to conduct this study to understand different adaptations to similar environmental conditions.